# Optical Diagnostics of the Maxillary Sinuses by Digital Diaphanoscopy Technology

**DOI:** 10.3390/diagnostics11010077

**Published:** 2021-01-06

**Authors:** Ekaterina O. Bryanskaya, Irina N. Novikova, Viktor V. Dremin, Roman Yu. Gneushev, Olga A. Bibikova, Andrey V. Dunaev, Viacheslav G. Artyushenko

**Affiliations:** 1R&D Center of Biomedical Photonics, Orel State University, Orel 302026, Russia; irina.makovik@gmail.com (I.N.N.); dremin_viktor@mail.ru (V.V.D.); tef312@yandex.ru (R.Y.G.); dunaev@bmecenter.ru (A.V.D.); 2College of Engineering and Physical Sciences, Aston University, Birmingham B4 7ET, UK; 3Art Photonics GmbH, 12489 Berlin, Germany; olga.a.bibikova@gmail.com (O.A.B.); sa@artphotonics.com (V.G.A.)

**Keywords:** optical diagnostics, digital diaphanoscopy, magnetic resonance imaging, paranasal sinuses, inflammatory diseases, Monte Carlo simulation

## Abstract

The work is devoted to the development of a scientific and technical basis for instrument implementation of a digital diaphanoscopy technology for the diagnosis of maxillary sinus inflammatory diseases taking into account the anatomical features of patients (differences in skin structure, skull bone thickness, and sinus size), the optical properties of exercised tissues, and the age and gender characteristics of patients. The technology is based on visualization and analysis of scattering patterns of low-intensity radiation as it passes through the maxillary sinuses. The article presents the experimental data obtained using the digital diaphanoscopy method and the results of numerical simulation of the optical radiation passage through the study area. The experimental setup has been modernized through the installation of a a device for controlling the LED applicator brightness. The approach proposed may have considerable promise for creating diagnostic criteria for various pathological changes and can be used to assess the differences in the optical and anatomical features of males and females.

## 1. Introduction

Sinusitis is a common disease with worldwide prevalence and one of the leading causes of antibiotic prescription [1]. In 2018, 28.9 million people in the United States reported a sinusitis diagnosis in the previous 12-month period, which accounted for 11.6% of the adult population [2]. In Europe, sinusitis affects 10.9% of the population [3]. Delay in diagnosis and treatment of sinusitis may cause serious effects such as immune sensitivities to medication, development of different complications, including intracranial complications (50% of deaths). Therefore, an accurate, painless, and timely diagnosis of maxillary sinus pathology is one of the key problems of modern otolaryngology. Diagnostic imaging techniques, such as radiography, computed tomography, magnetic resonance imaging, ultrasound diagnostics (including assessment of stiffness and echogenicity), and rhinoscopy, are important tools to detect this kind of disorder, but they are not recommended for pregnant women and children due to the use of carcinogenic roentgen radiation during the study, painfulness of the diagnostic procedures, and a high level of false-negative results.

Radiography is based on the high penetrating power of radiation and its absorption ability. It allows assessing the general condition of the paranasal sinuses and makes it possible to detect the presence of liquid content (along with formed cysts or polyps) in them, as well as the changes in the mucous membrane. If there is a pathological change in sinus pneumatization, then the resulting image will show significant darkening along the upper horizontal level, asymmetry of the inflamed sinuses, narrowing of the nasal passages, and mucosal thickening [4]. Despite the advantages of this technique, its application in clinical practice is limited to assessments that help to identify the degree of development of a pathological process which already has pronounced signs to determine further treatment tactics and the need for surgical interventions. It should also be noted that a high dose of radiation exposed to pregnant women raises the risk of hypertension, may slow down the growth of the fetus, and cause the worsening of the health indicators of newborns.

The major contribution of computed tomography (CT) is in providing additional useful information. The CT method, similar to other imaging techniques, uses X-ray radiation and its distribution in tissues depending on their density. CT provides the differentiation of soft tissues and can distinguish between density differences of no more than 0.1%. However, it is also associated with significant radiation exposure [5].

Magnetic resonance imaging (MRI), endoscopy (rhinoscopy), or ultrasound are used as alternative methods. The MRI method consists in registering the excitation of hydrogen atomic nuclei by a certain combination of electromagnetic waves in a constant magnetic field of high intensity [5]. MRI can be useful in assessing lesions in nasal bones, inflammatory processes in the paranasal sinuses, formed cysts or polyps, chronic diseases, and changes in bone structure resulting from injury. However, this method is characterized by a high level of false positive results, poor visualization of bone tissue, and is contraindicated in the presence of implants, prostheses, and obesity. The disadvantages of this method are the cost and high microwave load on the patient.

Rhinoscopy is performed by placing a flexible fiber-optic tube into the nasal passage to assess the color of the mucous membrane, its humidity, the shape of the nasal septum, the caliber of vessels, the condition of the nasal shell, and the size and content of the nasal passages. Although rhinoscopy yields important information and adequate sample collection, it also has some disadvantages which include its high cost and discomfort to the patient.

Ultrasound makes it possible to diagnose inflammatory processes in the sinuses, cysts, polyps, and traumatic damage to the walls of the paranasal sinuses, as well as to detect foreign bodies. Normally, the paranasal sinuses contain air, which serves as an obstacle to the propagation of ultrasonic waves. In the absence of pathological changes, the echo signal from the sinus is not detected due to its complete reflection from the air contained in it. If the walls of the sinuses are thickened, for example, due to edema of the mucosa, or they contain pathological contents, then there occur conditions for the propagation of ultrasound waves. When passing through the sinus, ultrasound waves are reflected from its posterior parts and are fixed by an ultrasound device. The results of ultrasound scanning do not always correspond to reality, which can be attributed to deficiencies in equipment design or incorrect interpretation of the results.

Non-invasive optical technologies are increasingly used in various fields of medicine to diagnose pathological conditions. The digital diaphanoscopy method has the potential to separate normal and pathological conditions (inflammation, cystic, and tumor tissues) of the maxillary sinuses. It is based on the visualization of scattering patterns of low-intensity radiation as it passes through the maxillary sinuses [6,7,8]. Although the method of diaphanoscopy or transillumination has a long history of usage [9,10], especially in ophthalmology [11] and urology [12], it is of limited use in otolaryngology. First examinations of tissue parameters in the near infrared were performed by Beuthan in 1982. Relying on these tests, infrared transillumination seems to be a promising tool for the rapid diagnosis of sinusitis, but white light spectrum radiation is a limiting factor influencing the application of this approach in otolaryngology [13,14]. White light does not provide a complete diagnostic picture because it is absorbed and dispersed by tissues to a great extent; no informative features and algorithms capable of separating normal and pathological conditions exist. At present, there are no clinically justified classifying features, developed classification models, and diagnostic criteria that can differentiate pathological changes in the maxillary sinuses (inflammation, cysts, and tumor tissues) by digital diaphanoscopy.

The purpose of this study was to develop a scientific and technical basis for instrument implementation of the technology proposed for diagnosing maxillary sinus pathology, taking into account the anatomical features of the study area (skin structure, bone thickness of the facial part of the skull, sinus size, and asymmetry), its optical properties, and age and gender characteristics.

## 2. Materials and Methods

### 2.1. Experimental Setup

For the realization of this approach, the experimental setup was designed and assembled (Figure 1a). Low-intensity radiation of the visible (650 nm) and NIR (850 nm) ranges was used for translucence of the paranasal sinuses. The CMOS-camera was applied to visualize the light scattering pattern [15]. To minimize the external illumination influence, a protective screen was designed which facilitated research under various external conditions. To suppress movement artifacts, the volunteers were asked to place their head on a chin rest.

The LED applicator takes into account the anatomical features of the study area, namely, the features of the oral cavity and maxillary sinuses (Figure 1b). During the diagnostic procedure, the applicator was placed in the oral cavity, and then the measurements were performed (Figure 1c). Before each use, the surface of the applicator was cleaned with wipes impregnated with a disinfecting solution. The solution composition included purified water, isopropanol, ethanol, didetsildimetilammonium chloride, and dodecyl dipropylene triamine.

The micro-LEDs OSRAM Opto Semiconductors GmbH (Regensburg, Germany) with wavelengths of 650 (C4L-H12T5) and 850 nm (F3453) and 8 pieces on the right and left sides for each wavelength were used as radiation sources for the applicator. The CMOS-camera UI-3240CP-NIR-GL Rev.2 IDS GmbH (Obersulm, Germany) and the lens Pentax С1614-M (Tokyo, Japan) were applied to register diagnostic information in the prototype device. The camera provides high-speed image acquisition, high quality, and maximum quantum efficiency (light sensitivity) in the selected spectral range. The control unit provides the LEDs sequential switching on and off at wavelengths of 650 and 850 nm, as well as their performance in the setting mode (LEDs flashing at a wavelength of 650 nm).

The specially designed software assists us in analyzing the obtained scattering pattern. It allows for trial control (selection of a radiation source and a camera operation mode), gathering of personal patient data, and recording of diaphanoscopic images and their quantitative (visualization) and qualitative processing (pseudo-color image segmentation). The software also provides the function of uploading a diaphanoscopic examination report, which contains information about the patient, diaphanoscopic images, diagnostic assessments, and quantitative values (the percentage of light passed through the sinus).

### 2.2. Study Design

The study groups were formed taking into account the differences in age, gender parameters, and anatomical features of the study area, as well as the existing pathological changes and inflammatory processes of the maxillary sinuses. Preliminary experimental studies were conducted in 20 conditionally healthy volunteers and 15 patients with suspected maxillary sinus inflammation at the medical diagnostic center “MediScan” (Orel, Russia).

The study was approved by the Ethics committee of the Orel State University (record of meeting No. 15 of 21 February 2019) and carried out in accordance with the 2013 Declaration of Helsinki by the World Medical Association. After receiving the description of the protocol, the volunteers signed an informed consent indicating their voluntary willingness to participate in the study.

The experiments were conducted under established protocols. The subjects were in a sitting position. After the pre-disinfected LED-applicator was inserted into the oral cavity of the subject, the subject head was placed in a precision positioning unit and, then, together with the CMOS-camera, it was covered with a protective screen. To account for the effect of the camera exposure time on the light scattering patterns, images were recorded at 40 different camera exposure times in the range from 0 to 40 ms (increment of 1 ms). In the patient study, the measurement results were additionally compared with the T2 weighted images by MRI studies, which were performed using the 1T MRI Scanner of the Magnetom series, Siemens (Munich, Germany).

## 3. Results

### 3.1. Results of Preliminary Experimental Studies

The data obtained in trials with healthy volunteers showed that changing the camera exposure time does not significantly affect the diagnostic result. Therefore, further analysis of the experimental data was carried out at the camera exposure time of 20.7 ms.

Figure 2 and Figure 3 present the registered and processed images for two conditionally healthy volunteers (a man and a woman of the same age group) with the same camera exposure time of 20.7 ms for probing radiation wavelengths of 650 nm (a) and 850 nm (b).

Preliminary experimental studies using the relevant method MRI have confirmed the sensitivity of the digital diaphanoscopy method in detecting pathological changes in the maxillary sinuses [6]. Figure 4 gives examples of the T2 weighted MRI images (Figure 4a) and the images registered and processed by diaphanoscopy (Figure 4b).

Analysis of the registered and processed images obtained by digital diaphanoscopy revealed that the cyst area is characterized by the lowest intensity compared to other structures, which can be explained by the strong absorbing properties of the cystic fluid in the near-infrared range [6]. The results of digital diaphanoscopy are determined by the optical properties of the study area [16,17,18,19,20,21,22,23] and their changes in various anatomical and gender features [24,25]. In our study, we applied Monte Carlo simulation to take into account the effect of the anatomical and gender characteristics of patients on the scattering pattern of light, to justify the medical and technical requirements for the instrument, and to adjust the parameters of the LED applicator.

### 3.2. Monte Carlo Simulation

Since the object of research has a rather complex organization, a simplified model of the maxillary sinus was developed to establish the regularity of the weakening of the probing signal from the anatomical and gender features of the studied area (differences in the skin structure, the thickness of the skull bone tissue, and the size of the sinuses). The Monte Carlo methodology was used for the construction of the 3D model. This method is one of the most effective simulation tools when dealing with biological tissues [26,27]. Figure 5 shows a scheme of the developed model.

In the model, the environment is represented by 8 main layers, as well as by an additional layer in the form of a pathological change (cystic fluid or tumor). The optical characteristics of the biological tissues used in the simulation are presented in Table 1.

The thickness and size of the layers and their absorption and scattering coefficients were set for both females and males. Since the sizes and thicknesses of the layers depend on gender and age [18,19], the layer thicknesses were averaged within one gender to simplify the developed model. The thicknesses of the simulated layers are given in Table 2.

Analysis of the optical properties of the research area indicates high absorbing properties of the hypodermis at the selected wavelengths of probing radiation. In addition, the results of the preliminary experimental studies demonstrate that the changes in the hypodermis thickness strongly affect the diagnostic result.

The Monte Carlo simulation involving a simplified model of the research area was performed for 650 and 850 nm radiation sources in the TracePro software environment (Lambda Research Corporation) [34,35,36]. The number of simulated photons was 10^6^. The power of probing radiation in the simulation for the wavelengths of 650 and 850 nm was 8 mW.

Figure 6, Figure 7 and Figure 8 show the simulation results of the probe radiation propagation (the photons path through the biological tissue and the irradiance map) for the maxillary sinus of female (a) and male (b) without pathology (Figure 6), with cystic fluid (Figure 7) and with tumor (Figure 8).

Figure 9 illustrates the difference in radiation power (intensity) reduction in males and females. This decrease has a more pronounced character when the pathology in the sinuses is observed in the NIR range (850 nm) and can be attributed to the optical features of pathological tissues, namely, the high absorption properties at selected wavelength [16,17].

Besides, the results demonstrate a decrease in the intensity of radiation at the detector (radiant power) when it passed through the biological tissues at different values of the bone tissue and skin, the sinus size [22,23].

The revealed regularity confirmed the results of the experimental studies. It was also found that the adjustment of the parameters of the probing and measuring parts of the device for implementation of the proposed technology is necessary to ensure similar scattering light patterns for different patients and their further comparison.

### 3.3. Upgrade of the Experimental Setup

Based on the simulation results, the experimental setup was upgraded; the block scheme of the setup is shown in Figure 10.

A controller of the LED applicator brightness was designed and installed in the setup in addition to the unit controlling the output power of the probing applicator. It is positioned in the gap between the LED control unit of the applicator and the LED applicator itself. To control the operation of the controller of the LED applicator, an additional software has been developed, which allows one to change the voltage supplied to the LEDs, as well as to measure the current flow in real time and to calculate the power consumption. The software makes it possible to save many brightness profiles and switch them immediately before starting the measurements, thereby automatically selecting the desired range of radiation power of the applicator for specific volunteers and patients in accordance with their anatomical features.

The experimental studies, which were conducted using the modernized installation, allowed the detection of changes in the power consumption of the LEDs applicator. To identify the values of power consumption specific to each patient based on their gender and anatomical features, the study involved conditionally healthy volunteers; the power consumption of the LEDs applicator varied from 0 to 750 mW in increments of 50 mW. The camera exposure time remained unchanged. It was established that, in healthy male volunteers, the maximum power consumption of the LEDs applicator was insufficient to obtain an adequate scattering pattern of light passing through the sinuses, which is associated with their anatomical features (bone thickness, skin, and size of the sinuses). In female volunteers, the maxillary sinuses were visualized in the range of LEDs power consumption equal to 300–500 mW.

The ranges of changes in the radiation flux for the two radiation sources were also revealed. Thus, it was found that at 850 nm the radiation flux varies in the range from 0 to 200 mW, whereas for the 650 nm radiation source this parameter changes in the range from 0 to 18 mW.

In the future, the elements of the controller of the LED applicator brightness will be adjusted and replaced, and a new applicator will be designed to increase the radiated light power.

## 4. Discussion

In this study, we tested a device designed to implement the digital diaphanoscopy technology, which is based on visualization and analysis of the low-intensity radiation scattering pattern in the maxillary sinuses.

The review and analysis of existing methods (CT and MRI) for the diagnosis of inflammatory diseases of maxillary sinuses diseases showed their limitation either for the repeated conduct of studies due to radiation or microwave exposure or for the conduct of studies in general, for example, for pregnant women or children. In otolaryngology, the standard methods for diagnosing such pathologies (ultrasound and rhinoscopy techniques) sometimes yield false positive results due to complexity in interpreting the results, or due to trauma-related aspects. In comparison with the considered methods, the method of digital diaphanoscopy allows one to overcome these drawbacks.

In addition, the review of the literature in the field of non-invasive optical diagnosis of paranasal pathology demonstrates that our technology has the advantage over the previously published results, as it provides a foundation for the assessement of the condition of the sinuses for all categories of patients, based on their anatomical and gender features. For this purpose, we designed the original brightness controller of the LED applicator and developed a specialized adjustment software for the probing mode, which makes it possible to select an effective radiation dose for each patient.

Currently, further experimental studies are being conducted to form an appropriate database and identify diagnostic criteria for various pathological changes, taking into account the range of the optical power of probing radiation, that have the greatest sensitivity to visualization of pathological changes in the maxillary sinuses in different study groups divided by gender.

## 5. Conclusions

Preliminary trials were conducted in 20 conditionally healthy volunteers and 15 patients with suspected maxillary sinus inflammation. The influence of anatomical and gender features of the study area on the diagnostic results (differences in skin structure, skull bone thickness, and sinus size) was revealed. The sensitivity of the prototype device to detect pathological changes was confirmed by the results of MRI studies.

The simulation results show the regularity of changes in the light scattering and parameters of the probing and measuring parts of the experimental setup. The mathematical model developed via Monte Carlo simulation made it possible to take into account the anatomical and gender features of the study area, as well as the absorption and scattering of optical radiation.

The prototype of the device was upgraded to obtain similar scattering patterns of light for different patients and to ensure their comparison. To adjust the output power of the probing applicator, a device for controlling the LED applicator brightness was designed, and additional software was developed.

The obtained results can be used to create modern diagnostic devices for the diagnosis of maxillary sinus pathology based on visualization and analysis of the low-intensity radiation scattering pattern. The application of the developed digital diaphanoscopy technology will make it possible to conduct timely, reliable, and painless diagnostics of maxillary sinus pathology, assess the dynamics of changes in the pathological processes within the framework of the therapy, and analyze its effectiveness. It is important to note that, due to the portability and simplicity of its instrument implementation, the technology can be used as a screening method for assessing the condition of the maxillary sinuses both in hospital and medical institutions and remotely in the absence of otolaryngologists and diagnosticians.

## Figures and Tables

**Figure 1 diagnostics-11-00077-f001:**
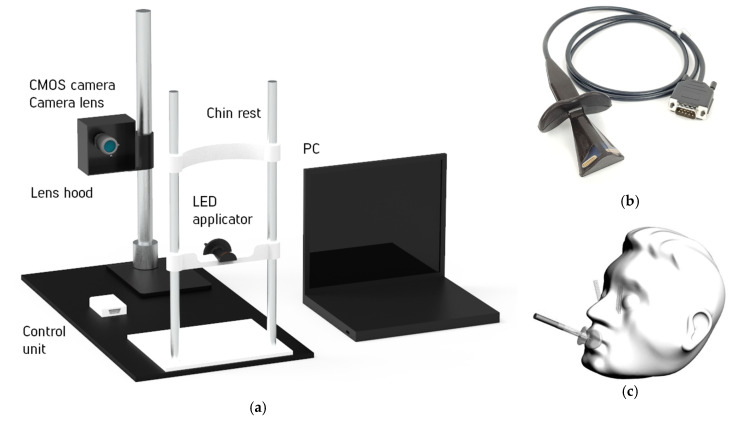
General view of the experimental setup (**a**); the LED applicator (the working part) (**b**); and the position of the applicator during measurement with the propagation of probing rays (3D model) (**c**).

**Figure 2 diagnostics-11-00077-f002:**
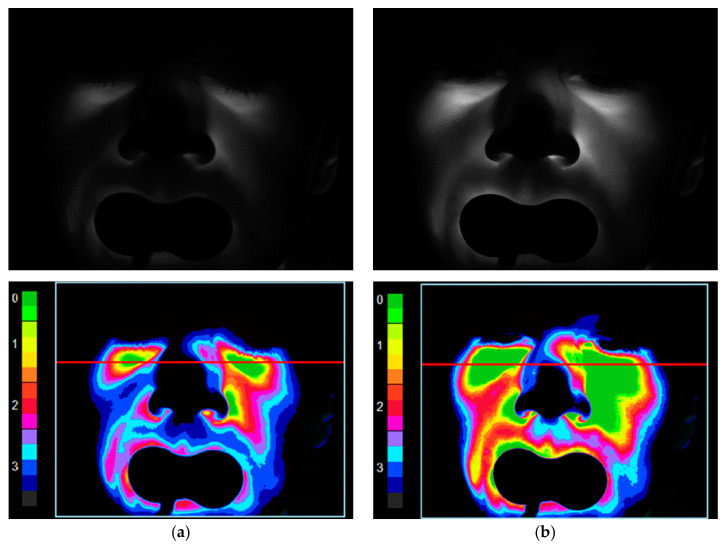
Registered (**top**) and processed (**bottom**) images for a conditionally healthy volunteer 1 (male) at a camera exposure time of 20.7 ms for probing radiation wavelengths of: 650 nm (**a**); and 850 nm (**b**). The red line is the selection of the area for analyzing the transmitted light.

**Figure 3 diagnostics-11-00077-f003:**
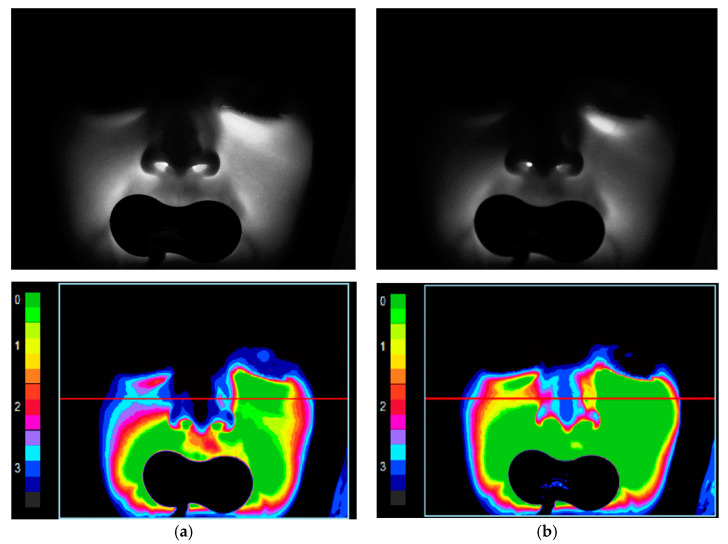
Registered (**top**) and processed (**bottom**) images for a conditionally healthy volunteer 2 (female) at a camera exposure time of 20.7 ms for probing radiation wavelengths of: 650 nm (**a**); and 850 nm (**b**).

**Figure 4 diagnostics-11-00077-f004:**
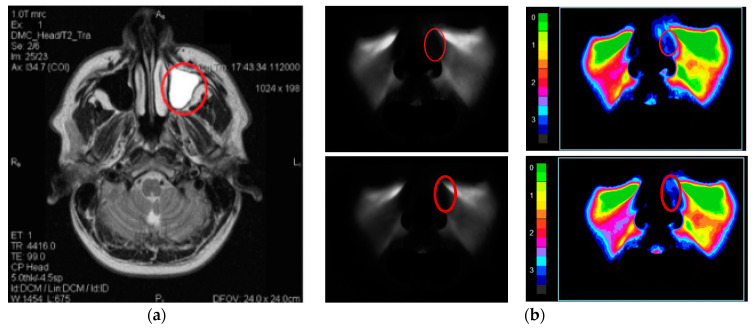
The T2 weighted MRI image (**a**); and the images registered and processed by diaphanoscopy (**b**) for a patient (male) at radiation wavelengths of 650 nm (**top**) and 850 nm (**bottom**).

**Figure 5 diagnostics-11-00077-f005:**
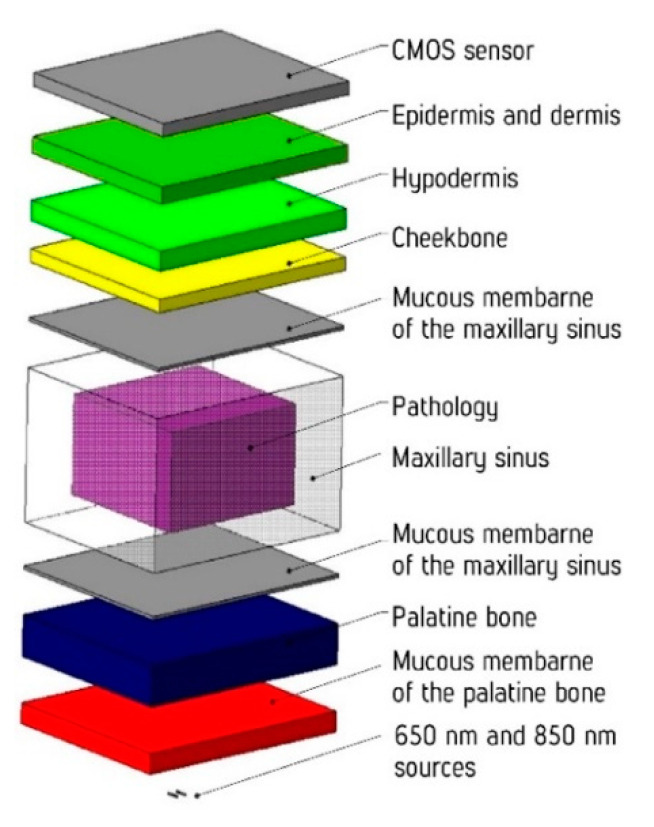
Full 3D view of the developed model.

**Figure 6 diagnostics-11-00077-f006:**
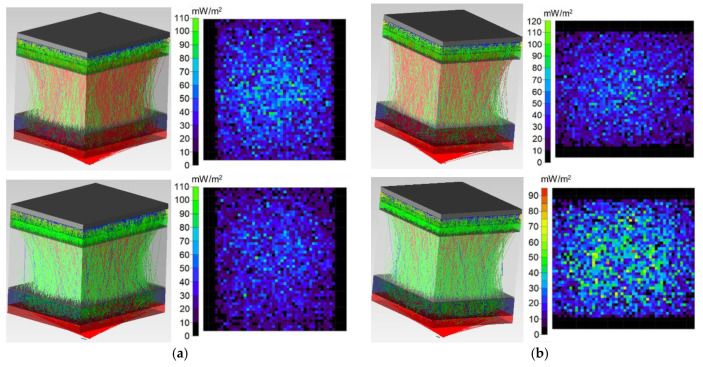
Simulation results for the probe radiation propagation through the maxillary sinus of female (**a**) and male (**b**) without pathology at a wavelength of 650 nm (**top**) and 850 nm (**bottom**).

**Figure 7 diagnostics-11-00077-f007:**
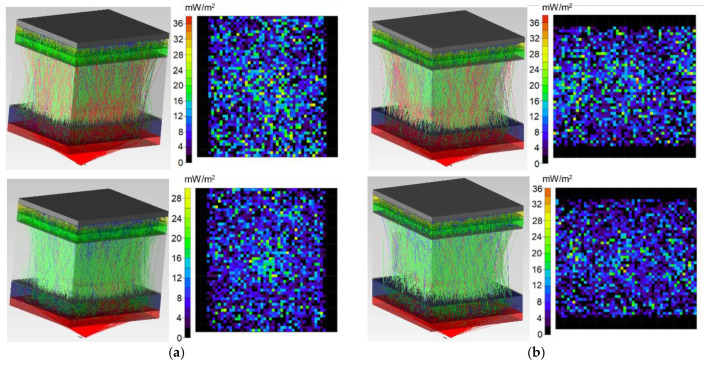
Simulation results for the probe radiation propagation through the maxillary sinus of female (**a**) and male (**b**) with cystic fluid at a wavelength of 650 nm (**top**) and 850 nm (**bottom**).

**Figure 8 diagnostics-11-00077-f008:**
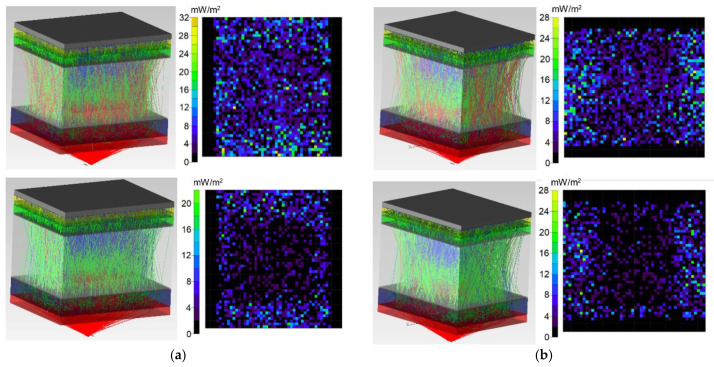
Simulation results for the probe radiation propagation through the maxillary sinus of female (**a**) and males (**b**) with tumor at a wavelength of 650 nm (**top**) and 850 nm (**bottom**).

**Figure 9 diagnostics-11-00077-f009:**
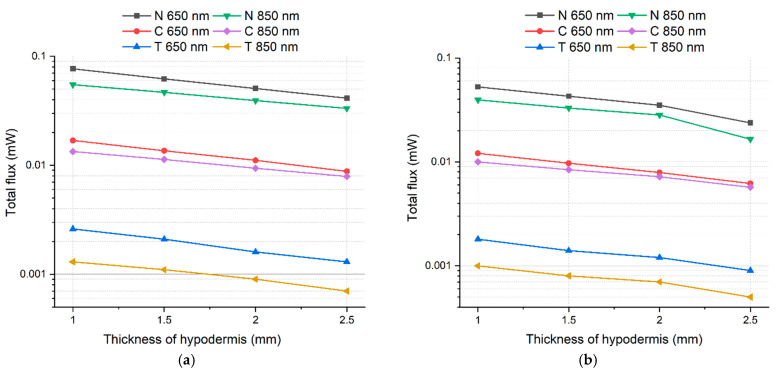
Dependence of the change in the total flux (power) of radiation coming to the camera detector on the change in the hypodermis thickness and on the presence of pathology in the sinuses of female (**a**) and males (**b**) for wavelengths of 650 and 850 nm. The following labels are used: “N” for healthy tissues, “C” for tissues with cyst, and “T” for tissues with tumor.

**Figure 10 diagnostics-11-00077-f010:**
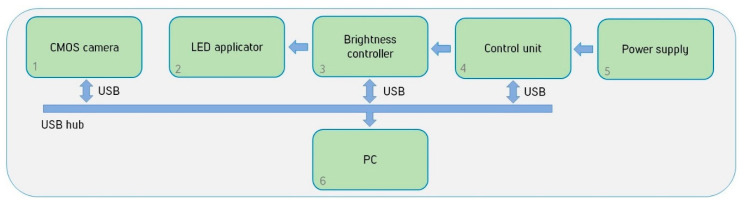
The block scheme of a modernized experimental setup.

**Table 1 diagnostics-11-00077-t001:** The optical characteristics of biological tissues.

Biological Tissue Layer	Wavelength λ, nm	Absorption Coefficient μ_a_, mm^−1^	Scattering Coefficient μ_s_, mm^−1^
Mucous membrane (sinus/palatine bone) [18]	650 850	0.05 0.075	0.8 1.2
Zygomatic/Palatine bone [19,20]	650 850	0.011 0.007	1.873 2.113
Cystic fluid [16,17]	650 850	0.022 0.027	1.34 0.95
Tumor [21]	650 850	0.0391 0.0522	2.17 2.67
Hypodermis [22]	650 850	0.18 0.1	2 2.7
Epidermis and dermis [23]	650 850	0.17 0.2	3 3.7

**Table 2 diagnostics-11-00077-t002:** The thickness of the simulated layers, mm.

Layer	Male	Female
Mucous membrane of a palatine bone [25,28]	2	3
Palatine bone [24,25,29]	3.1	3.1
Mucous membrane of a sinus [17]	0.5	0.5
Sinus [29,30]	6.7	6.2
Zygomatic bone [31,32]	26	23
Hypodermis [33]	3	3
Epidermis and dermis [33]	1.5–3	1–2

## Data Availability

The data that support the findings of this study are available upon reasonable request request from the corresponding author.

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
