# Peer review of "Optical Diagnostics of the Maxillary Sinuses by Digital Diaphanoscopy Technology"

_diagnostics, 2021, doi:10.3390/diagnostics11010077_

Round 1

Reviewer 1 Report

The research objective is novel and may be highly interesting mainly in a clinical perspective providing an important objective tool for the diagnosis to assess the maxillary sinus. The manuscript presents only a few flaws as described below:

- following the PICOT+ guidelines, the Title should provide more information on the participant involved in the study, on time and study design;

- the study design should be clarified at the beginning of this section;

- it is not specified where and how the patients were recruited;

- the sample size and the power of the study are completely missing. If not possible, the lack of sample size pre-determination may be discussed as a limitation;

- some details on the methodology used may be added, i.e. types of data collected for each patient, clinical chart;

- the initials and above all qualifications of the operators performing each step of the study may be stated;

- a list of abbreviation used in the study should be provided.

Author Response

Response to reviewers comments

We would like to express our appreciation to the Reviewers for their valuable questions, remarks, and comments. As a consequence, we have thoroughly revised the manuscript according to the needed changes and provide below our point-by-point replies to the comments. Reviewers’ original comments are written below in Italic font. The changes made in the manuscript text are marked in red.

Following the PICOT+ guidelines, the Title should provide more information on the participant involved in the study, on time and study design;

Response: Thank you. To comply with the PICOT+ principles, the abstract has been modified to correctly display the research question.

Changes in the text: In Abstract: The work is devoted to the development of a scientific and technical basis for instrument implementation of a digital diaphanoscopy technology for the diagnosis of maxillary sinus inflammatory diseases taking into account the anatomical features of patients (differences in skin structure, skull bone thickness, and sinus size), the optical properties of exercised tissues, and the  age and gender characteristics of patients. The technology is based on visualization and analysis of scattering patterns of low-intensity radiation as it passes through the maxillary sinuses. The article presents the experimental data obtained using the digital diaphanoscopy method and the results of numerical simulation of the optical radiation passage through the study area. The experimental setup has been modernized through the installation of a a device for controlling the LED applicator brightness. The approach proposed may have considerable promise for creating diagnostic criterions for various pathological changes and can be used to assess the differences in the optical and anatomical features of males and females.

The study design should be clarified at the beginning of this section;

Response: Thank you for your comment. We have added detailed information in the section 2.2. Study Design.

Changes in the text:  In Section 2.2. Study Design: The study groups were formed taking into account the differences in age, gender parameters, and anatomical features of the study area, as well as the existing pathological changes and inflammatory processes of the maxillary sinuses. Preliminary experimental studies were conducted in 20 conditionally healthy volunteers and 15 patients with suspected maxillary sinus inflammation at the medical diagnostic center «MediScan» (Orel, Russia).

The study was approved by the Ethics committee of the Orel State University (record of meeting No. 15 of 21.02.2019) and carried out in accordance with the 2013 Declaration of Helsinki by the World Medical Association. After receiving the description of the protocol, the volunteers signed an informed consent indicating their voluntary willingness to participate in the study.

The experiments were conducted under established protocols. The subjects were in a sitting position. After the pre-disinfected LED-applicator was inserted into the oral cavity of the subject, the subject head was placed in a precision positioning unit and, then, together with the CMOS-camera, it was covered with a protective screen. To account for the effect of the camera exposure time on the light scattering patterns, images were recorded at 40 different camera exposure times in the range from 0 to 40 ms (increment of 1 ms). In the patient study, the measurement results were additionally compared with the T2 weighted images by MRI studies, which were performed using the 1T MRI Scanner of the Magnetom series (Siemens, Germany).

It is not specified where and how the patients were recruited;

Response: Thank you for your comment. Detailed information is provided in the new section 2.2. Study Design.

The sample size and the power of the study are completely missing. If not possible, the lack of sample size pre-determination may be discussed as a limitation;

Response: Thank you. At the moment, the sample size was set to 20 conditionally healthy volunteers and 15 patients with suspected maxillary sinus inflammation. In addition, in the new discussion section, we noted that we will expand the database.

Some details on the methodology used may be added, i.e. types of data collected for each patient, clinical chart;

Response: Thank you. We have added this information to the section 2.1. Experimental Setup.

Changes in the text: In Section 2.1. Experimental Setup: The specially designed software assists us in analyzing the obtained scattering pattern. It allows for trial control (selection of a radiation source and a camera operation mode), gathering of personal patient data, and recording of diaphanoscopic images and their quantitative (visualization) and qualitative processing (pseudo-color image segmentation). The software also provides the function of uploading a diaphanoscopic examination report, which contains information about the patient, diaphanoscopic images, diagnostic assessments, and quantitative values (the percentage of light passed through the sinus).

The initials and above all qualifications of the operators performing each step of the study may be stated;

Response: Thank you for your comment. In the section Author Contributions, it is noted which part of the work was performed by each of the authors.

A list of abbreviation used in the study should be provided.

Response: Thank you for your comment. We have added a list of abbreviations to the manuscript text.

Changes in the text:

List of abbreviations:

CT – computed tomography

MRI – magnetic resonance imaging

NIR – near infrared

CMOS – Complementary Metal Oxide Semiconductor

LED – Light-emitting diode

Reviewer 2 Report

Dear Autohrs,

The article has an interesting topic but English and spelling errors should be reviewed

Abstract: the first paragraph must be changed. It should be an introduction to the topic and not mention results.

Introduction: It must refer to the muscular ultrasound evaluation through echogenicity as a practice both in terms of characterization and diagnosis

Methods: Must have subtopics with participants characterization; how were the participants chosen; the informed consent; how did the authors used the experimental setup in the participants.

The study was not submitted to an ethics committee? if yes, the document number must be presented.

Discussion: It should be more exhaustive in terms of comparing the results with the literature. The authors must have a subtopic with limitations of the study.

Conclusion: Must present future ideas and how can it be used in clinical practice.

Best regards,

Author Response

Response to reviewers comments

We would like to express our appreciation to the Reviewers for their valuable questions, remarks, and comments. As a consequence, we have thoroughly revised the manuscript according to the needed changes and provide below our point-by-point replies to the comments. Reviewers’ original comments are written below in Italic font. The changes made in the manuscript text are marked in red.

The article has an interesting topic but English and spelling errors should be reviewed.

Response: The text of the manuscript has been improved. The revisions introduced when polishing English in the manuscript have not altered the main content, and therefore we have not marked small changes in the revised text.

Abstract: the first paragraph must be changed. It should be an introduction to the topic and not mention results.

Response: Thank you. Appropriate changes have been introduced into the abstract.

Changes in the text: In Abstract: The work is devoted to the development of a scientific and technical basis for instrument implementation of a digital diaphanoscopy technology for the diagnosis of maxillary sinus inflammatory diseases taking into account the anatomical features of patients (differences in skin structure, skull bone thickness, and sinus size), the optical properties of exercised tissues, and the  age and gender characteristics of patients. The technology is based on visualization and analysis of scattering patterns of low-intensity radiation as it passes through the maxillary sinuses. The article presents the experimental data obtained using the digital diaphanoscopy method and the results of numerical simulation of the optical radiation passage through the study area. The experimental setup has been modernized through the installation of a a device for controlling the LED applicator brightness. The approach proposed may have considerable promise for creating diagnostic criterions for various pathological changes and can be used to assess the differences in the optical and anatomical features of males and females.

Introduction: It must refer to the muscular ultrasound evaluation through echogenicity as a practice both in terms of characterization and diagnosis.

Response: In the Introduction section, we describe the main possibilities of ultrasound diagnostics. We clarified that this method is also used to assess the stiffness and echogenicity of tissues.

Methods: Must have subtopics with participants characterization; how were the participants chosen; the informed consent; how did the authors used the experimental setup in the participants.

Response: Thank you for your comment. Appropriate changes have been introduced into the manuscript text.

Changes in the text: In Section 2.2. Study Design: The study groups were formed taking into account the differences in age, gender parameters, and anatomical features of the study area, as well as the existing pathological changes and inflammatory processes of the maxillary sinuses. Preliminary experimental studies were conducted in 20 conditionally healthy volunteers and 15 patients with suspected maxillary sinus inflammation at the medical diagnostic center «MediScan» (Orel, Russia).

The study was approved by the Ethics committee of the Orel State University (record of meeting No. 15 of 21.02.2019) and carried out in accordance with the 2013 Declaration of Helsinki by the World Medical Association. After receiving the description of the protocol, the volunteers signed an informed consent indicating their voluntary willingness to participate in the study.

The experiments were conducted under established protocols. The subjects were in a sitting position. After the pre-disinfected LED-applicator was inserted into the oral cavity of the subject, the subject head was placed in a precision positioning unit and, then, together with the CMOS-camera, it was covered with a protective screen. To account for the effect of the camera exposure time on the light scattering patterns, images were recorded at 40 different camera exposure times in the range from 0 to 40 ms (increment of 1 ms). In the patient study, the measurement results were additionally compared with the T2 weighted images by MRI studies, which were performed using the 1T MRI Scanner of the Magnetom series (Siemens, Germany).

The study was not submitted to an ethics committee? if yes, the document number must be presented.

Response: Thank you for your question. Appropriate changes have been introduced into the manuscript text.

Changes in the text: In Section 2.2. Study Design: The study was approved by the Ethics committee of the Orel State University (record of meeting No. 15 of 21.02.2019) and carried out in accordance with the 2013 Declaration of Helsinki by the World Medical Association.

Discussion: It should be more exhaustive in terms of comparing the results with the literature. The authors must have a subtopic with limitations of the study.

Response: We thank the Reviewer for this question. Appropriate changes have been introduced into the manuscript text. We have created a separate Discussion section where we have also included the limitations and future directions of this study.

Changes in the text: In Section 5. Discussion: In this study, we have tested a device designed to implement the digital diaphanoscopy technology, which is based on visualization and analysis of the low-intensity radiation scattering pattern in the maxillary sinuses.

The review and analysis of existing methods (CT and MRI) for the diagnosis of inflammatory diseases of maxillary sinuses diseases showed their limitation either for the repeated conduct of studies due to radiation or microwave exposure, or for the conduct of studies in general, for example, for pregnant women or children. In otolaryngology, the standard methods for diagnosing such pathologies (ultrasound and rhinoscopy techniques) sometimes yield false positive results due to complexity in interpreting the results, or due to trauma-related aspects. In comparison with the considered methods, the method of digital diaphanoscopy allows one to overcome these drawbacks.

In addition, the review of the literature in the field of non-invasive optical diagnosis of paranasal pathology demonstrates that our technology has the advantage over the previously published results, as it provides a foundation for the assessement of the condition of the sinuses for all categories of patients, based on their anatomical and gender features. For this purpose, we have designed the original brightness controller of the LED applicator and developed a specialized adjustment software for the probing mode, which makes it possible to select an effective radiation dose for each patient.

Currently, further experimental studies are being conducted to form an appropriate database and identify diagnostic criteria for various pathological changes, taking into account the range of the optical power of probing radiation, that have the greatest sensitivity to visualization of pathological changes in the maxillary sinuses in different study groups divided by gender.

Conclusion: Must present future ideas and how can it be used in clinical practice.

Response: Thank you. Appropriate changes have been introduced into the Conclusion text.

Changes in the text: In Section 6. Conclusion: Preliminary trials were conducted in 20 conditionally healthy volunteers and 15 patients with suspected maxillary sinus inflammation. The influence of anatomical and gender features of the study area on the diagnostic results (differences in skin structure, skull bone thickness, and sinus size) was revealed. The sensitivity of the prototype device to detect pathological changes was confirmed by the results of MRI studies.

The simulation results showed the regularity of changes in the light scattering and parameters of the probing and measuring parts of the experimental setup. The mathematical model developed via Monte Carlo simulation made it possible to take into account the anatomical and gender features of the study area, as well as the absorption and scattering of optical radiation.

The prototype of the device was upgraded to obtain similar scattering patterns of light for different patients and to ensure their comparison. To adjust the output power of the probing applicator, a device for controlling the LED applicator brightness was designed, and additional software was developed.

The obtained results can be used to create modern diagnostic devices for the diagnosis of maxillary sinus pathology based on visualization and analysis of the low-intensity radiation scattering pattern. The application of the developed digital diaphanoscopy technology will make it possible to conduct timely, reliable, and painless diagnostics of maxillary sinus pathology, assess the dynamics of changes in the pathological processes within the framework of the therapy and analyze its effectiveness. It is important to note that due to the portability and simplicity of its instrument implementation, the technology can be used as a screening method for assessing the condition of the maxillary sinuses both in the hospital in medical institutions, and remotely in the absence of otolaryngologists and diagnosticians.